# Factors Related to Weight Loss Maintenance in the Medium–Long Term after Bariatric Surgery: A Review

**DOI:** 10.3390/jcm10081739

**Published:** 2021-04-16

**Authors:** Isabel Cornejo-Pareja, María Molina-Vega, Ana María Gómez-Pérez, Miguel Damas-Fuentes, Francisco J. Tinahones

**Affiliations:** 1Department of Endocrinology and Nutrition, Virgen de la Victoria University Hospital, 29010 Málaga, Spain; isabelmaria_cornejo@hotmail.com (I.C.-P.); migueldamasf@hotmail.com (M.D.-F.); fjtinahones@hotmail.com (F.J.T.); 2Instituto de Investigación Biomédica de Málaga (IBIMA), Virgen de la Victoria University Hospital, 29010 Málaga, Spain; 3Centro de Investigación Biomédica en Red de la Fisiopatología de la Obesidad y Nutrición, Instituto de Salud Carlos III, 28029 Madrid, Spain

**Keywords:** bariatric surgery, weight regain, surgical technique, psychological disorders, physical activity, diet, gut hormones, gut–brain axis

## Abstract

Despite bariatric surgery being the most effective treatment for obesity, some individuals do not respond adequately, especially in the long term. Identifying the predictors of correct weight maintenance in the medium (from 1 to 3 years after surgery) and long term (from 3 years and above) is of vital importance to reduce failure after bariatric surgery; therefore, we summarize the evidence about certain factors, among which we highlight surgical technique, psychological factors, physical activity, adherence to diet, gastrointestinal hormones or neurological factors related to appetite control. We conducted a search in PubMed focused on the last five years (2015–2021). Main findings are as follows: despite Roux-en-Y gastric bypass being more effective in the long term, sleeve gastrectomy shows a more beneficial effectiveness–complications balance; pre-surgical psychological and behavioral evaluation along with post-surgical treatment improve long-term surgical outcomes; physical activity programs after bariatric surgery, in addition to continuous and comprehensive care interventions regarding diet habits, improve weight loss maintenance, but it is necessary to improve adherence; the impact of bariatric surgery on the gut–brain axis seems to influence weight maintenance. In conclusion, although interesting findings exist, the evidence is contradictory in some places, and long-term clinical trials are necessary to draw more robust conclusions.

## 1. Introduction

Obesity is defined as the pathological increase in adipose tissue associated with chronic low-grade inflammation and an increased risk of many pathological conditions such as type 2 diabetes mellitus (T2DM), cardiovascular disease, or cancer [1,2]. It is considered an epidemic disease and is expected to affect 44% of the adult population of the USA in 2031 and 31% of the adult population of Europe in 2037 [3].

The first-line treatment for obesity is lifestyle intervention, including a healthy diet and physical activity to produce a negative energy balance [2]. In those patients with moderate-risk or high-risk obesity, pharmacological therapy is indicated [2]. A weight loss of 5–10% can be easily attained and maintained for a time by lifestyle modification programs and anti-obesity medications. However, the weight usually recovers progressively from the first year after the intervention onwards [4].

Bariatric surgery is the most effective treatment for weight loss and weight-loss maintenance. Weight loss with bariatric surgery can reach 50–75% of excess body weight (EBW) and can be maintained 10 years later [4]. Nevertheless, the efficacy of bariatric surgery is not uniform between patients, with some of them not obtaining satisfactory weight loss from the beginning (primary non-responders) or regaining weight in the long term (secondary non-responders) [5].

In this review, we will focus on the factors that influence weight loss in the medium–long term after bariatric surgery.

## 2. Surgical Technique

Since Edward Mason reported effective weight loss after the first gastric bypass in the mid-1960s, many bariatric procedures such as jejunoileal bypass, vertical banded gastroplasty, and laparoscopic adjustable gastric band (LAGB) have been used and later abandoned because of adverse events or inadequate long-term efficacy [6,7]. In 1994, the first report of the use of the laparoscopic technique was a landmark in bariatric surgical care as laparoscopic surgery reduces postoperative pain, time recovery, wound infection, and late ventral hernia formation in comparison to conventional techniques [6]. Nowadays, the most frequently performed bariatric procedures are laparoscopic Roux-en-Y gastric bypass (RYGB) and, especially, laparoscopic sleeve gastrectomy (SG), which accounts for 61% of primary bariatric procedures in the USA [7]. Therefore, we are going to analyze the evidence (clinical trials and meta-analysis) in the last 5 years comparing RYGB and SG regarding weight loss in the medium–long term.

In a recent meta-analysis, including 7443 patients from 23 studies, Hu et al. [8] found that there was no difference in excess weight loss (EWL)% between RYGB and SG in the short term (3 months–2 years), but RYGB was superior to SG in the mid-term (3 years) and long term (5 years) after surgery. However, RYBG showed more late complications than SG. Previously, Yang et al. [9] found similar results in a meta-analysis of 15 randomized controlled trials (1381 patients), concluding that SG and RYBG were similar regarding weight loss at <3 years but that EWL% was greater with RYBG 5 years after surgery, although with a higher incidence of complications. Other smaller meta-analyses found comparable results [10,11]. Likewise, King et al. [12] reported that weight regain seems to be higher after SG in comparison to RYGB. Conversely, other authors have concluded that there is no difference in weight loss between SG and RYBG at 1 year [13,14] and at 3 years [14,15] after surgery. Data from 10 or more years show that RYGB is able to maintain substantial weight loss, but data on SG are insufficient for a meta-analysis [16].

Results from clinical trials comparing weight loss between SG and RYGB published in the last 5 years are compiled in Table 1. 

The Sleeve vs. Bypass (SLEEVEPASS) trial was a multicenter, multisurgeon, open-label randomized trial whose main aim was to determine if SG and RYGB were equivalent for weight loss in 240 patients. At five years since surgery, it was observed that EWL% after SG was 49% and after RYGB 57%, and this difference was not statistically significant despite that higher weight loss was achieved with RYGB [17]. Similar results were maintained at 7 years after surgery [18]. Regarding obesity co-morbidities, SG and RYGB were similar in T2DM remission and dyslipidemia resolution, where RYGB was better than SG in hypertension resolution at 5 years [17]. At 1 year after surgery, both Hofsø et al. [19] and Murphy et al. [20] reported RYGB to be superior to SG for weight loss (total weight loss 29% vs. 23%; *p* < 0.001 and EWL% 84.2% vs. 70.2%, *p* = 0.002, respectively). However, the primary outcome of these trials was T2DM remission, not weight loss, and although Hofsø et al. [19] found a higher remission of T2DM in RYGB in comparison to SG, Murphy et al. [20] observed both surgical procedures to be similar. In a small clinical trial, Schneider et al. [21] reported a higher EBMIL% (excess body mass index loss) with RYGB in comparison to SG (76.4% vs. 64.4%, *p* = 0.046) after 17 ± 5.6 months of follow-up. However, they also compared both surgical techniques regarding body composition and resting energy expenditure, not finding significant differences. In the Swiss Multicenter Bypass or Sleeve Study (SM-BOSS) trial, Peterli et al. [22,23] reported similar EBMIL% when comparing SG and RYGB at 1, 2, 3, and 5 years after surgery. Similarly, no statistically significant differences were observed between one anastomosis gastric bypass and SG at 1 year [24] and 3 years [25] after surgery. On the contrary, as that reported in the meta-analysis performed by Hu et al. [8], Ignat et al. [26] showed that, although EWL% was similar between SG and RYGB in the short term, a higher EWL% was achieved with RYBG vs. SG in the medium-term (at 3 years: 83% vs. 66.3%, *p* = 0.024) and long-term (at 5 years: 74.8% vs. 65.1%, *p* = 0.045) follow-up.

In summary, despite many studies concluding that SG and RYGB are comparable at weight loss in the medium and long term, other studies have found RYGB to be better than SG regarding this outcome and also in obesity-related co-morbidities (such as T2DM or hypertension, between others) resolution. However, SG seems to produce fewer complications than RYGB. Maybe new clinical trials [27] will be able to tip the balance in favor of RYBG or confirm the equivalence of both surgical procedures in weight loss at medium and long term.

## 3. Psychological Factors

Psychological difficulties and poorly treated mental health can negatively affect the results of bariatric surgery [28]. Mood, emotional dysregulation, depression, poor health literacy, and deficits in executive functioning, attention, and memory skills, among others, are likely to be important barriers to effective maintenance of weight loss [29], consistently finding deficiencies of these skills in the obese population compared to lean people [30].

A growing body of evidence suggests that deficits in executive function are common in obesity [31,32], finding a constant inverse association between obesity and executive function in children, adolescents, and the adult population [33].

Obese subjects show a pronounced impairment in decision-making and real-life learning in terms of reward and punishment (by the Iowa gambling task (IGT)) [34], and impaired central coherence (processing style centered on the details) that makes it impossible for them to see the “big picture” in a similar way to patients with anorexia nervosa [35,36]. In addition, the obese subject is impulsive and has poor performance on tests of global cognitive function and memory [37]. These deficits in executive function are considered the cause of inappropriate attitudes towards food and represent a trigger for both eating disorders and changes in BMI [38]. Likewise, obese individuals show an unregulated physiological response to intense emotion by tending to increase their food intake during periods of emotional arousal and/or stress, a response known as emotional eating [39]. However, the nature of this obesity-associated cognitive decline is unclear. Different explanations have been proposed including factors driven by inflammation, dopamine dysregulation implicated in hyperphagia, vascular diseases or neuroendocrine changes in ghrelin and leptin [40,41,42]. 

### 3.1. Cognitive Impairment

The presence of cognitive impairment in the obese subject can be particularly problematic in the population undergoing bariatric surgery, given the many lifestyle changes required after it. Up to 23% of subjects undergoing bariatric surgery have clinically significant cognitive impairment, and approximately 40% have more subtle cognitive deficits [43]. Such deficiencies in executive function have been associated with maladaptive eating behaviors, including uncontrolled or uninhibited eating along with sedentary behaviors, and may contribute to suboptimal weight loss after bariatric surgery [44] Spitznagel et al. found that preoperative baseline cognitive impairment predicted the outcome of weight at one year after bariatric surgery (RYGB) in 84 obese individuals. Poorer initial cognitive function in the domains of executive ability, attention, and memory predicted a lower percentage of weight loss and higher BMI at 12 months after bariatric surgery. Impairments in memory or executive function could interfere with the patient’s ability to plan and follow postoperative guidelines for successful maintenance of weight loss [45]. Furthermore, cognition has been shown to improve shortly after bariatric surgery [46], and this initial improvement appears to be of substantial importance in its predictive ability for sustained weight loss. Supporting this notion, Spitznagel and colleagues [47,48] found that early postoperative cognitive dysfunction (at 12 postoperative weeks) predicted progression at 24 and 36 months. Poorer performance on cognitive tests at 12 weeks (lower performance in executive ability, attention, and memory) was indicative of a reduction in weight loss at 2- and 3-year follow-up after bariatric surgery. In this sense, Alosco et al. [49] evaluated 50 obese subjects who underwent RYGB, finding early cognitive benefits (12 weeks) that were generally maintained up to 36 months after surgery. Interestingly, it was observed in this work that the reduction in the domain of attention 24-36 months after the intervention was associated with weight recovery in this time. Kulendran et al. [50] in a study with 45 patients found that impulsivity measured as an inhibitory control of executive function together with the type of surgery (most effective RYGB vs. SG) were able to predict weight loss 6 months after bariatric surgery. The results found regarding the relationship between weight loss and executive performance in bariatric surgery may suggest that a reduction in body fat favors an improvement in executive function as a consequence of the resolution of metabolic alterations related to obesity. Likewise, a lower cognitive deficit at the beginning would lead to improvements in eating habits linked to a greater reduction in BMI, as we have seen. Similarly, cognitive function seems to be related to the durability of weight loss after bariatric surgery [47,48]. The cognitive skills that seem to best predict the results of weight loss included memory (particularly recognition memory) and executive functions (specifically working memory and generativity), and adherence behaviors could be the likely mechanism by which cognitive dysfunction leads to poorer performance in reducing long-term weight loss in bariatric surgery [33]. However, Bergh et al. [51], after evaluating 230 who underwent RYGB, found that while certain psychological factors such as self-esteem, planning, disposition to change behavior, or depressive symptoms, among others, were related to postoperative adherence to dietary recommendations and physical exercise. However, no associations were found in relation to weight loss one year after surgery. 

### 3.2. Eating Disorders

Another important point in the failure of weight loss after bariatric surgery is related to the presence of eating disorders (EDs). Recent studies [52,53] have reported a higher prevalence of EDs among patients undergoing bariatric surgery with weight regain, with binge eating disorder especially prevalent in this population [54]. Conceição et al., [53] in a longitudinal study, found that up to 65% of patients who experienced weight regain between 17 and 20 months after surgery (both LAGB or RYGB) suffered from ED postoperatively. Furthermore, other studies have emphasized the role of other ED such as emotional eating, night eating syndrome (NES), or picking and nibling (P&N) in the results of bariatric surgery and how they contribute to suboptimal weight loss [55]. 

### 3.3. Depression

The reciprocal, longitudinal link between depression and obesity has been demonstrated in different studies [56]. Nevertheless, the exact nature of the relationship between depression and maintenance of obesity remains unclear, perhaps because clinical depression is a common exclusion criterion in weight loss intervention trials [57]. 

A recent meta-analysis [58] provides evidence for bariatric surgery, finding a reduction in depression symptoms at 6, 12, and 24 months after surgery. However, these symptoms increased after 36 months in a similar way to the baseline situation. Similar studies showed that improvements in depressive symptoms after bariatric surgery may not be maintained after 1–3 years after surgery, worsening again as in the starting point [59]. Weight regain and depression after surgery can act as a mutual risk factor. A depressed mood is associated with unhealthy lifestyle habits, emotional eating and loss of eating control [60], and weight regain after bariatric surgery [58,60,61]. Novelli et al. [62] found a higher score on emotional eating in obese women who underwent RYGB with insufficient weight loss 2 years after surgery. Feig et al. [63], in a cross-sectional study of 95 subjects undergoing RYGB and SG mainly, suggested that positive psychological states (positive affect or optimism) could be relevant in the state of well-being after bariatric surgery, finding greater adherence to healthy behaviors, physical activity, and weight loss. However, these associations lost statistical significance when factors such as depression were included.

### 3.4. Impulsive Behavior

Loss-of-control (LOC) eating is a common characteristic among subjects undergoing bariatric surgery [64], especially widespread in the adolescent population [65], and is associated with poorer weight outcomes. Goldschmidt et al. [64,66] and White et al. [67] determined that postoperative LOC eating constitutes a phenotype that negatively affects the weight result, being prospectively related to greater long-term weight recovery after RYGB, while pre-surgical eating LOC was not related to changes in post-surgery BMI. The rates of LOC eating decreased in the period immediately after surgery (6 months) compared to baseline; however, these rates increased gradually over time (2–4 years) after surgery.

### 3.5. Other Psychological Factors

It has been investigated whether different personality types predict the results of weight after bariatric surgery, without being able to draw clear conclusions. While some showed no influence in this regard [68], Gordon et al. found that they could influence the amount of weight loss at 2 years of RYGB [69]. 

In conclusion, multiple physicological factors are related to weight loss after bariatric surgery. An integrative and multiple approach that includes pre-surgical psychological and behavioral evaluation along with post-surgical treatment can be corrective for weight regain and persistence of obesity. In addition, addressing depression and executive deficits before and after bariatric surgery is needed to improve long-term surgical outcomes. Future research should further explore the best way to consider cognitive deficits in preoperative detection and follow-up of candidates for bariatric surgery.

## 4. Physical Activity

National Institute for Health and Care Excellence (NICE) [70] recommends that the postoperative follow-up of the obese patient should incorporate counseling and support for physical activity.

### 4.1. Lack of Adherence to Exercise Training in Bariatric Surgery Patients

People with severe obesity can, generally, safely exercise vigorously [71]; however, candidates for bariatric surgery are generally less active than normal-weight subjects [72]. Additionally, candidates for bariatric surgery are more sedentary than the general obese population. Likewise, of all postoperative recommendations, those related to physical activity are commonly the most non-compliant [73]. King et al. [74] examined the physical activity of 310 patients who underwent bariatric surgery through the use of accelerometers, finding that most of the subjects increased their level of physical activity 1 year after bariatric surgery (RYGB mainly and other techniques included such as LAGB, SG, banded gastric bypass, or biliopancreatic diversion with duodenal switch) compared to baseline. However, most remained with poor physical activity according to the American Diabetes Association and the American College of Sports Medicine (<150 min per week), and some even decreased their activity compared to baseline. Bond et al. [75] compared self-reported estimates of physical activity vs. those based on objective measurements by an accelerometer in 20 patients who underwent bariatric surgery (65% LAGB and 35% RYGB) 6 months after surgery. Although in the postoperative period 55% of the participants self-reported adherence to the physical activity recommendations, only 5% were objectified by accelerometer measurement, with the changes in physical activity of moderate to vigorous intensity being much smaller than the self-reported. Ouellette et al. [76] also found no changes in early postoperative physical activity compared to baseline in subjects undergoing bariatric surgery (29% RYGB and 71% SG), with no correlation between the levels of physical activity self-reported by the patient and those observed by accelerometry. In addition, participants failed to adhere to the minimum recommended physical activity (150 min per week of moderate to vigorous intensity physical activity). Taken together, these data suggest that a high proportion of patients after bariatric surgery do not increase their physical activity, and some even decrease it, identifying a relevant area of intervention.

### 4.2. Aerobic and Resistance Training

Increased physical activity has been associated with greater weight loss after bariatric surgery [77,78,79,80,81,82,83,84]. Furthermore, close supervision and monitoring of exercise programs support greater weight loss compared to minimally supervised programs [85]. Egberts et al. [78] in a systematic review of observational studies (on 3852 patients) found a relationship between increased physical exercise (measured by physical activity questionnaires) and weight loss after bariatric surgery (LAGB and RYBG). In addition, the meta-analysis showed an average of 3.62 kg of greater weight loss with the practice of physical activity. 

#### 4.2.1. Aerobic Training

Carnero et al. [77] in a study carried out on 96 patients who underwent bariatric surgery (RYGB), monitored physical activity and effects on weight and body composition according to a 6 month structured exercise program, observing greater weight loss and more favorable body composition (less fat mass and greater muscle mass) in patients who performed moderate physical activity and decreased sedentary time. Furthermore, patients in the highest quartiles of physical activity achieved greater reductions in adiposity, reporting a dose–response association between exercise time and adiposity, already revealed by previous studies. In this sense, Woodlief et al. [86] demonstrated that patients who performed a greater amount of exercise (286 ± 40 min per week) after RYGB were those who obtained the greatest loss of weight and body fat compared to those who performed less physical activity. However, other studies have not supported this finding [87,88,89,90]. Coen et al. [88] examined the efficacy of a physical exercise program (120 min/week of treadmill walking for 6 months) in severely obese subjects, not observing any additional impact on RYGB-induced weight loss or fat mass. These findings are similar to those of Shah et al. [89] who showed how the prescription of a high-volume exercise program (energy expenditure in exercise > 2000 Kcal/week) with bariatric surgery (70% GB and 30% RYGB) at least 3 months earlier had no impact on the body weight or circumference of waist compared to the control group. The lack of effect of exercise on weight in these studies is probably due to the strong initial influence of surgery; thus, these data do not rule out the possibility that an exercise program may cause additional weight loss and improve body composition or adiposity favorably after surgery. Furthermore, after the initial large loss, weight tends to stabilize, and the long-term sustainability of this weight loss is probably more related to lifestyle changes such as avoiding sedentary behavior and regular physical activity [90]. 

#### 4.2.2. Combination of Aerobic and Resistance Training

A randomized clinical trial introducing a 12-week structured and supervised physical exercise program in 24 post-bariatric surgery (surgical technique not specified) patients (at 12-24 months later) and 12 controls with the same characteristics demonstrated improvements in capacity/physical function and weight, among other parameters [91]. In this sense, Rothwell and colleagues [79] reported that weight loss after a semi-structured exercise program at 12 months of bariatric surgery (LAGB) improved, without observing this effect at 36 months. Hanvold and colleagues [81] found that patients undergoing RYGB who reported physical activity ≥150 min/week had a lower percentage of weight regain compared to less active participants. However, they found no differences when comparing the diet and physical activity-focused lifestyle intervention group vs. the usual care group at long-term (2 years). Coleman et al. [90] found that a structured post-bariatric exercise program improves the physical capacity of patients (strength, balance, flexibility, mobility, coordination) at 6–24 months post-surgery (GS, RYGB, LAGB), without finding additional effects on weight loss. 

A recent meta-analysis [80] of 15 exercise training studies (aerobic training in 5 studies, resistance training in 2 studies, and a combination of aerobic and resistance training in 8 studies) also concluded that physical training programs carried out after bariatric surgery (RYGB and SB mainly) were effective in optimizing the loss of weight and fat mass and improving the physical condition of the patients, although no additional effect on lean mass loss was described.

### 4.3. Maintenance of Muscle Mass

The maintenance of muscle mass is vital to optimize physical functioning and preserve energy expenditure at rest. The latter represents 60–70% of total energy expenditure [70], finding greater reductions and less recovery of visceral abdominal fat when it is included physical exercise in weight loss programs [92]. Loss of fat free mass (FFM) can predispose to long-term weight regain. Metcalf et al. [93] found that duodenal switch surgery in patients adhering to an exercise program (30 min per session, with > 3 sessions a week) achieved 28% more loss of fat mass and 8% more gain of lean mass compared to sedentary patients at 18 months postoperatively. A systematic review by Chaston et al. [94] suggests that loss of FFM (skeletal muscle, bone, and organs) represents a weight percentage of 31.3% of weight loss after RYGB. Although the significance of the loss of this FFM is not well known, excessive loss may be undesirable. Specifically, in older patients, the loss of muscle mass and bone mineral density may have a negative impact on their physical function, sarcopenia, and quality of life [95]. Physical exercise, and specifically endurance exercise, is effective in maintaining muscle mass [96].

In summary, despite that physical activity programs after bariatric surgery have been shown to be associated with a higher weight loss and a more beneficial body composition, most patients do not increase, and may even decrease, physical activity. However, most of the papers refer to the early postoperative stages, and the evidence is very limited in the long term. More interventional clinical trials with long-term structured exercise programs are needed to determine whether exercise is important in preventing weight regain in bariatric surgery patients.

## 5. Dietary Factors

In the bariatric population in particular, the diet is often poor, and caloric intake often increases progressively after bariatric surgery [97]. Sawer et al. [98] found an increase in caloric intake 2 years after bariatric surgery compared to the first 5 months (1172.9 ± 46.5 Kcal/day vs. 1358.1 ± 60.5 Kcal/day), finding greater weight loss and maintenance in those with greater dietary adherence.

In the National Weight Control Registry (NWCR), a large-scale prospective study to investigate the maintenance of long-term weight loss, among the dietary strategies adopted for the stable maintenance of weight loss, the following stand out: Adherence to a low-calorie and low-fat diet, eating breakfast regularly, and maintaining a consistent eating pattern throughout the week [99]. However, the literature on dietary advice to improve weight after bariatric surgery is limited. In addition, the studies in this regard present a small sample size, as well as heterogeneity of dietary support, settings, times, duration, type of surgery, etc.

The main macronutrients in food (carbohydrates, proteins, and fats) stimulate oxygen consumption in different ways, which can influence changes in body weight and possibly subsequent weight regain. Bray et al. [100] in the POUNDS LOST Study and Grave et al. [101] found no effect of diet composition on body weight or energy expenditure [100]. However, Reid et al. [102] found higher carbohydrate and alcohol consumption in those subjects who had regained weight after an average of 12 years since bariatric surgery, compared to those who had maintained weight loss. Frequent consumption of high-fat and high-sugar snacks can lead to excessive energy intake from carbohydrates, and this behavior may reduce the maintenance of weight loss [103]. Restricting the consumption of soft drinks or carbonated beverages is another important aspect that has been related to the stability of postsurgical weight [104]. Likewise, different studies [105,106] have found that a diet high in protein and with a low glycemic index was the best option to maintain weight loss, and this macronutrient composition could be related to a lower decrease in energy expenditure in the subjects who followed it [105].

Regarding dietary behavior, and more specifically behaviors related to reduced rations and frequency of intake, they have been related to more favorable weight 3 years after bariatric surgery [107]. Similar findings have been described in a cohort of 50 adolescents undergoing bariatric surgery [87].

It is likely that numerous mechanisms contribute to changes in lifestyle after bariatric surgery. Continuous and comprehensive care interventions appear to be the most successful approaches to maintaining weight loss. However, more long-term randomized clinical trials are needed to clarify these issues.

## 6. Gut Hormones and Neuronal Factors

### 6.1. Gut Hormones

Bariatric surgery produces changes in gastrointestinal anatomy and functionality that may be implicated in different ways in weight loss after the procedure and weight maintenance in the long-term. Regulation of appetite and eating is a complex process that depends on the integration of signals from the digestive tract to the central nervous system (CNS). Specifically, there are regions in the hypothalamus and brainstem that integrate peripheral signals to coordinate orexigenic and anorexigenic responses. Those signals provide information about energy availability depending on nutritional state and energy storage in adipose tissue. There is a very intricate system of signals between the gut, vagal afferents, hypothalamus, brainstem, and reward centers in response to nutrient ingestion to regulate energy homeostasis [108].

The main gut hormones implicated in energy homeostasis are ghrelin, which is orexigenic, peptide tyrosin-tyrosin (PYY), glucagon-like peptide 1 (GLP-1), oxyntomodulin (OXM), glicentin, pancreatic polypeptide (PP), amylin and cholecystokinin (CCK), which are anorexigenic [109]. Ghrelin increases appetite and food intake, accelerates gastric emptying, increases gastric acid secretion, decreases insulin secretion, and stimulates hepatic glucose production. Its levels are higher just before nutrient intake, and there is a ghrelin suppression after a meal; this suppression is greater following a high-carbohydrate meal compared to a high-fat meal [108]. On the contrary, PYY reduces food intake and appetite, increases insulin secretion, and delays gastric emptying. The peak in PYY secretion takes place typically 15–30 min after food intake, and protein and fat-rich foods stimulate greater peaks of this hormone compared to carbohydrates [110]. GLP-1 has a biphasic secretion after nutrients intake with an early phase 15 min after ingestion and a second peak at 30–60 min [111]. Its effects are similar to PYY—suppressing appetite, reducing food intake, and delaying gastric emptying—but it also promotes glucose-dependent insulin secretion [112]. OXM is co-secreted with GLP-1 in response to food ingestion, and it reduces energy intake, increases energy expenditure related to physical activity, delays gastric emptying, and also stimulates glucose-dependent insulin secretion. Glicentin seems to have a role in stimulating insulin secretion, and decreasing gut motility and acid secretion in animals, but its biological role is not fully elucidated yet. PP is secreted after nutrients ingestion depending on caloric load, and its main functions are the inhibition of gastric emptying, pancreatic exocrine secretion, and gallbladder motility. Amylin levels reach a peak one hour after nutrient ingestion and remain high for four hours, slowing gastric emptying, suppressing glucagon postprandial secretion, inhibiting energy intake, and increasing energy expenditure [108]. Finally, CCK promotes gallbladder contraction and pancreatic exocrine secretion favoring food digestion, but it also slows gastric emptying, inhibits acid gastric secretion, decreases energy intake, and stimulates insulin secretion [113].

Some alterations in the normal function of these hormones have been reported in obese patients compared to lean subjects, also in syndromic obesity. Some changes in these hormones have been identified following different weight loss strategies, including bariatric surgery (Table 2). For example, an increase in postprandial levels of GLP-1 in patients after SG and RYGB has been reported in several studies, and this change may persist in the long-term (at least 1–2 years). In the case of GIP (Gastric inhibitory polypeptide), data are more controversial, as some studies have reported an increase in postprandial levels after bariatric surgery, but some others did not find any change, especially after RYGB. Similar effects have been observed in a lot of studies for OXM and PYY, with increases in postprandial levels after RYGB and for PYY also after SG. Regarding ghrelin, the only orexigenic gut hormone of those previously mentioned, its suppression is usually improved after bariatric surgery. However, the mechanism seems to be different depending on the technique, as in RYGB the effect observed is in postprandial ghrelin, and in SG the effect observed is in fasting ghrelin. Although this is controversy, lower ghrelin levels may have beneficial effects on appetite regulation and in body weight [114].

Differential behaviors of gut hormones depending on surgical technique could be related with anatomical changes (duodenum exclusion in RYGB and restriction of the gastric fundus in SG) produced in the surgery and with different exposure to carbohydrates and fat. Based on this idea, there are three hypothesis that try to explain weight control. The hindgut hypothesis poses that the accelerated delivery of nutrients to the distal gut increases insulinotropic signals that are mediated, among others, by GLP-1, and that improves postprandial glucose and free fatty acids metabolism, favoring body weight control [115]. Besides, nutrient delivery to the distal intestine also may produce an increase in intestinal gluconeogenesis and may activate a hepato-portal sensor that leads to neural signals for reduced food intake and decreased glucose output from the liver, as the midgut hypothesis proposes [116]. Finally, foregut hypothesis suggests that bypassing the duodenum may reduce some factors that induce insulin resistance and β-cell dysfunction, decreasing diabetogenic signals [117].

Among the studies published in last five years regarding gut hormones and weight maintenance after bariatric surgery, there are some interesting results, though the majority of them are observational studies. Perakakis et al. performed two independent trials to assess circulating levels of gut hormones in response to different types of bariatric surgery and its influence on weight loss after a year of follow up. They compared the fasting and postprandial levels of nine gut hormones after a mixed meal test, before and after bariatric surgery (laparoscopic gastric banding, SG and RYGB), and they related them with weight loss, looking for predictors of long-term weight loss. Their most robust results referred to OXM and glicentin, which showed a significant increase 3 months after the surgery (SG and RYGB) that was maintained at one year. The percentage of weight change was related to this increase at 6 months (OXM: *p* = 0.004; glicentin: *p* = 0.001) and at 12 months (OXM: *p* = 0.053; glicentin: *p* = 0.049). For GLP-1, changes were more profound and significant for SG than for RYGB, in contrast with other studies. For GIP they only found a decrease after RYGB and no changes for SG, and finally for ghrelin there was a significant decrease after SG but no changes after RYGB. They concluded that glicentin increase may predict weight loss at 12 months better than GLP-1, and these effects seemed to be related with better satiety control [118]. In another comparative study, Santo et al. compared postprandial secretion of ghrelin, GIP, GLP-1, and leptin in patients with maintenance of more than 50% of the EWL (group A) versus patients with regain of more than 50% of the EWL (group B), with a follow up of 26 months. Although the sample size was very small and all patients had undergone RYGB, they found some interesting results. There was a decrease in postprandial ghrelin levels in both groups, suggesting better appetite control. GIP showed a relatively larger increase (with respect to baseline) in postprandial levels at 30 min in group A compared to group B (*p* = 0.01), and GLP-1 also showed a greater increase at 30 min in group A compared to group B (*p* = 0.05) as well as a greater relative increase with respect to baseline (*p* = 0.01). Finally, leptin showed greater basal levels in group B compared to group A (*p* = 0.02), suggesting that energetic reserves could have been larger in group B. Thus, they concluded that the increase in GLP-1 and GIP after nutrient intake may show the influence of these hormones in weight maintenance after RYGB [119]. Another similar prospective observational study led by Alamuddin analyzed postprandial GLP-1, PYY, ghrelin, and leptin levels at 6 and 18 months after bariatric surgery (SG and RYGB), and they compared with a control group. Despite the low number of patients who completed the 18 months of follow up, the results are interesting to understand gut hormone changes in the long term. They reported a decrease in fasting ghrelin levels, especially in the SG group at 6 months (*p* = 0.0199) and 18 months (*p* = 0.0003), and an exaggerated postprandial increase in GLP-1 and PYY at 6 months (RYGB: *p* < 0.0001; SG: *p* = 0.006) that lasted until 18 months only for GLP-1 [120]. With some differences, the results of these studies suggest that the increase in anorexigenic hormones levels and the decrease in orexigenic hormones may be related to weight loss and weight maintenance after bariatric surgery in the short and long-term.

#### Other Hormonal Factors

On the other hand, bile acids also may have a role in weight loss and weight maintenance after bariatric surgery. A lower increase in circulating levels of postprandial bile acids has been reported in obese individuals, and this fact may play a role in energetic metabolism and weight control because they have some hormonal effects, and they stimulate brown adipose tissue activity for thermogenic effects. Some studies have shown an increase in postprandial bile acids levels after RYGB, and this increase seems to be grater in the long-term. The exact mechanism is not known, but it could be related to the nutrient delivery to the distal small intestine [121]. In addition, the alterations in gut microbiota after RYGB could have a role because microbiota are a key regulator of bile acids conjugation and secondary bile acids formation [122]. Bile acid fasting levels correlate with GLP-1 peak levels and stimulate GLP-1 secretion, probably contributing to satiety and ß-cell insulin secretion [123]. Insulin secretion also may be facilitated via farnesoid X receptor (FXR), which directly responds to bile acid increase [124]. Moreover, there are bile acid receptors (TGR5 receptors) in skeletal muscle and brown adipose tissue. Thus, the binding of bile acids to these receptors may increase energy expenditure, facilitating thyroid hormone action. However, data are controversial, and it is not clear if energy expenditure contributes to weight maintenance after bariatric surgery [125].

### 6.2. Neuronal Factors

Several studies have suggested that changes in taste preferences after bariatric surgery, especially after RYGB and alterations in the reward system, may have an influence in weight maintenance after surgical treatment of obesity, though data are inconclusive [126].

The mesolimbic reward pathway is a dopaminergic pathway that is key in substance abuse disorders, and there is evidence that it is also important in obesity. Although food intake regulation is a very complex system with many actors implied, dopamine may mediate some aspects of eating behaviors. It is known that food reward and dopamine functions are altered in obesity [127]. Bariatric surgery (SG and RYGB) may increase striatal dopamine transmission, improving reward sensitivity. This improved sensitivity, along with other factors, may help to modify eating behaviors enhancing the preference for non-highly stimulating food. These changes in striatal dopamine transmission seem to be related to changes in gut hormone levels after bariatric surgery, such as the decrease in ghrelin levels and the increase in GLP-1 or PYY levels. Gut hormones are key in the connection of the gut and brain as well as microbiota, as some gut bacteria are also implicated in dopamine release, and the shift in gut microbiota after bariatric procedures may improve dopaminergic signaling. This microbiota–gut–brain axis is an important regulator of weight control, including after bariatric surgery [126].

Finally, another interesting point is the connection between appetite, taste preferences, and eating behavior since some changes in appetite and taste preferences have been reported after bariatric surgery [128]. A recent study by Zhang et al. [129] investigated the association between presurgical taste preferences and postsurgical weight regain. They included patients who underwent RYGB or SG and had at least 2 years of follow up, and they assessed preoperative taste preferences with a multichoice questionnaire. They found that patients with sweet food preferences had 5.5 kg of weight regain (*p* = 0.038), and patients with salty food preferences had 6.1 kg of weight regain (*p* = 0.048) compared to patients with no taste preferences. After adjustment, patients with salty food preferences showed the greater weight regain with 6.8 kg (*p* = 0.027) compared to patients with no preferences. Though these results from just one study do not allow to establish robust evidence, it is a very interesting approach to identify more factors related to weight maintenance in the long-term after bariatric surgery.

In summary, there are very complex and intricate systems connecting gut hormones, microbiota, and the CNS that play an important role in appetite control and energy homeostasis. Thus, the changes produced by SG and RYGB in this complex gut–brain axis seem to influence weight maintenance in the medium- and long-term after surgery.

## 7. Conclusions

Bariatric surgery is the most effective intervention for weight loss in obese patients, although it is not exempt from possible long-term failure and weight regain. Multiple factors that may be related to long-term weight maintenance have been described, ranging from the surgical technique itself to anatomical and functional modifications that lead to changes in the microbiota–gut–brain axis through gastrointestinal hormones, bile acids, and FXR-TGR5 influence on skeletal muscle and brown adipose tissue or dopaminergic pathways related to appetite control and energy homeostasis. Similarly, factors such as changes in lifestyle related to diet and physical activity, psychological factors such as executive function disorders, and the coexistence of depressive symptoms and eating disorders can play important roles in maintaining long-term weight loss. Therefore, numerous mechanisms may contribute to changes in lifestyle and weight maintenance after bariatric surgery; thus, continuous and comprehensive care interventions appear to be the most successful approaches to maintaining. However, some data are discordant, and more long-term studies are necessary in order to clearly identify predictive factors of weight regain that allow us to optimize the management and follow-up of the obese patient undergoing bariatric surgery.

## Figures and Tables

**Table 1 jcm-10-01739-t001:** Comparative clinical trials: SG vs. RY/OAGB.

	Sample Size (*n*)	Clinical Characteristics	Follow Up	Weight Loss (SG vs. RY/OAGB)	Conclusion
**Grönroos [18]** **2020**	SG = 121 RYGB = 119	Female sex (%): SG: 71.9, RYGB: 67.2 Mean age (years): SG: 48.5, RYGB: 48.4 T2DM (%): SG: 52, RYGB: 49	7 years	EWL%: 47 vs. 55	GB = SG
**Salminen [17]** **2018**			5 years	EWL%: 49 vs. 57	GB = SG
**Hofsø [19]** **2019**	SG = 55 RYGB = 54	Female sex (%): SG: 58, RYGB: 74 Mean age (years): SG: 47.1, RYGB: 48.2 T2DM (%): 100	1 year	TWL%: 23 vs. 29	GB > SG
**Murphy [20]** **2018**	SG = 58 RYGB = 56	Female sex (%): SG: 45, RYGB: 59 Mean age (years): SG: 45.5, RYGB: 46.6 T2DM (%): 100	1 year	EWL%: 70.2 vs. 84.2	GB > SG
**Shivakumar [25]** **2018**	SG = 100 OAGB = 101	Female sex (%): SG: 65, RYGB: 61.4 Mean age (years): SG: 39.9, RYGB: 42.9 T2DM (%): SG: 47, RYGB: 49	3 years	EWL%: 61.1 vs. 66.5	GB = SG
**Seetharamaiah [24]** **2016**			1 year	EWL%: 63.9 vs. 66.8	GB = SG
**Ignat [26]** **2017**	SG = 55 RYGB = 45	Female sex (%): SG: 78.2, RYGB: 86.7 Mean age (years): SG: 35.1, RYGB: 32.2 T2DM (%): NR	1 year	EWL%: 83 vs. 80.4	GB = SG
			2 years	EWL%: 77.8 vs. 79.8	GB = SG
			3 years	EWL%: 66.3 vs. 83	GB > SG
			5 years	EWL%: 65.1 vs. 74.8	GB > SG
**Peterli [22]** **2017**	SG = 107 RYGB = 110	Female sex (%): SG: 72, RYGB: 72 Mean age (years): SG: 43, RYGB: 42.1 T2DM (%): SG: 24, RYGB: 26	1 year	EBMIL%: 72 vs. 75	GB = SG
			2 years	EBMIL%: 75 vs. 78	GB = SG
			3 years	EBMIL%: 71 vs. 73	GB = SG
**Peterli [23]** **2018**			5 years	EBMIL%: 61.1 vs. 68.3	GB = SG
**Schneider [21]** **2016**	SG = 23 RYGB = 19	Female sex (%): SG: 87, RYGB: 84.2 Mean age (years): SG: 41.2, RYGB: 40.3 T2DM (%): SG: 57, RYGB: 42	17 months	EBMIL%: 64.4 vs. 76.4	GB > SG

SG: sleeve gastrectomy; RYGB: Roux-en-Y gastric bypass; OAGB: one anastomosis gastric bypass; T2DM: type 2 diabetes mellitus; EWL: excess weight loss; TWL: total weight loss; EBMIL: excess body mass index loss. GB > SG: gastric bypass better than sleeve gastrectomy; GB = SG: gastric bypass similar to sleeve gastrectomy.

**Table 2 jcm-10-01739-t002:** Summary of main changes in gut hormones after RYGB and SG.

Hormones	SG	RYGB
GLP-1	Increase	Increase
GIP	Increase/no changes	Increase/no changes
OXM	Increase	Increase
PYY	Increase	Increase
Glicentin	Increase	Increase
Ghrelin *	Suppression	Suppression

* All hormone levels refer to postprandial levels, except for ghrelin, whose changes occurred mainly in fasting levels. RYGB: Roux-en-Y gastric bypass. SG: sleeve gastrectomy. GLP-1: glucagon polypeptide like 1. GIP: glucose-dependent insulinotropic peptide OXM: oxyntomodulin. PYY: polypeptide tyrosine-tyrosine.

## Data Availability

Not applicable.

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
