# Peer review of "Factors Related to Weight Loss Maintenance in the Medium–Long Term after Bariatric Surgery: A Review"

_jcm, 2021, doi:10.3390/jcm10081739_

Round 1
Reviewer 1 Report
The manuscript entitled "Factors related to the weight loss maintenance in the medium long term after bariatric surgery: a review" is a narrative review, incorporating data on factors that are related to the effect of bariatric surgery (either sleeve gastrectomy or gastric bypass) and/or affected by such procedures in medium and long term follow up (3 years and 5 years respectively?). The review is written in a detailed manner and covers a wide spectrum of aspects and overall written in a comprehensive manner.
I would like to raise few comments regarding this manuscript:
- Major:
- The authors mention that the study reviews factors that are related to the surgery in the medium and long term, however, they do not mention what is long term and what is medium-term. It should be stated in the abstract and following the aim of the study.
- In that sense -if the long term means >=5 years and medium >=3 years (as mentioned in the section on "surgical techniques" ) then they should eliminate some data presented along with the text such as the studies mentioned in Table 1 – some include follow up for 1 year only. Also, along with the text, there are studies quoted that do not include information on follow-up time. In case some of them have less than 3 years follow up – they should be ignored unless the study aim is changed without time limitation….
- The authors mention on page 1 line 44 that they will focus on factors that influence weight loss during bariatric surgery (which ?) in the medium and long term (how long?). However, in the review, they move from mentioning different outcomes of bariatric surgery to outcomes related to weight loss achieved. I think that focusing only on factors that were suggested to explain weight-loss accomplished in gastric bypass vs sleeve gastrectomy after >= 3-5 years would make this review more focused and clear with a more "sharp message".
- Though the review is very informative, in some cases it is somehow confusing. I suggest re-organizing each section, so it is easier to follow and come up to conclusions or be able to come up with some "take-home messages". If each section would have been divided into two subsections – one describing information on gastric bypass and one on sleeve gastrectomy + each subsection is further divided according to outcomes (for example in the case of physical activity – from aerobic to resistance training to their combination with regard to weight loss success.
- Some studies quoted along the text do not include relevant information. For example, when authors mention a result of "bariatric surgery" they should mention which procedure (and preferably for how long was the follow-up). Example on page 7 -row 288: what kind of bariatric surgery? Here also the follow-up is 1 year…
- Minor:
- Abstract – there is no integrative summary of the main results discussed in the paper. The authors should include some directions and main findings stated in each section.
- Page 1 row 36 is an incomplete sentence.
- It is legitimate to focus on studies from the last five years, but it can be helpful to mention why were the past 5 years selected.
Author Response
Dear Editor and Reviewers,
We would like to thank you very much for your constructive comments and suggestions which have undoubtedly helped us to improve our manuscript.
In this new version of the manuscript, we have taken into consideration the comments and suggestions from reviewers and have revised the paper accordingly. We have responded to each of the reviewers' comments and have edited the manuscript to address all the reviewers' issues and suggestions.
We have provided the replies to the comments in the following section and have highlighted changes in the manuscript in blue.
We hope that our revised manuscript may now be found acceptable for publication in the journal. Nevertheless, we are of course willing to revise it further according to any other suggestions or concerns raised by the Editor or the Reviewers.
Yours faithfully,
María Molina-Vega, Ana María Gómez-Pérez, and coauthors
Reviewer Comments to Authors
Reviewer 1
The manuscript entitled "Factors related to the weight loss maintenance in the medium long term after bariatric surgery: a review" is a narrative review, incorporating data on factors that are related to the effect of bariatric surgery (either sleeve gastrectomy or gastric bypass) and/or affected by such procedures in medium and long term follow up (3 years and 5 years respectively?). The review is written in a detailed manner and covers a wide spectrum of aspects and overall written in a comprehensive manner.
I would like to raise few comments regarding this manuscript:
Response: First of all, we thank the reviewer for his/her comments and for offering us a constructive review of our manuscript
Major:
Comment 1: The authors mention that the study reviews factors that are related to the surgery in the medium and long term, however, they do not mention what is long term and what is medium-term. It should be stated in the abstract and following the aim of the study.
Response: Thanks for your pertinent commentary.
On one hand, most of the weight loss after bariatric surgery happens within the first year, but the loss of clinical response is often observed from this moment [Baig et al]. On the other hand, although studies analyzing the effect of surgical techniques show data at a term as long as 3 or 5 years after surgery, available data regarding gut hormones or neuronal factors reach shorter follow-up periods, we decided to consider results from at least 1 year since surgery, excluding those with less than 1 year of follow-up, in order to unify criteria for literature search. We have clarified this in the Abstract as follows: “Identifying the predictors of correct weight maintenance in the medium (from 1 to 3 years after surgery) and long term (from 3 years and above) is of vital importance to reduce failure after bariatric surgery”
Baig SJ, Priya P, Mahawar KK, Shah S; Indian Bariatric Surgery Outcome Reporting (IBSOR) Group. Weight Regain After Bariatric Surgery-A Multicentre Study of 9617 Patients from Indian Bariatric Surgery Outcome Reporting Group. Obes Surg. 2019 May;29(5):1583-1592. doi: 10.1007/s11695-019-03734-6.
Comment 2: In that sense -if the long term means >=5 years and medium >=3 years (as mentioned in the section on "surgical techniques" ) then they should eliminate some data presented along with the text such as the studies mentioned in Table 1 – some include follow up for 1 year only. Also, along with the text, there are studies quoted that do not include information on follow-up time. In case some of them have less than 3 years follow up – they should be ignored unless the study aim is changed without time limitation….
Response: Thanks for your commentary.
We would like to remark that the sentence “RYGB was superior to SG in the mid-term (3 years) and long-term (5 years) after surgery” is referred to data shown in a specific meta-analysis [Hu et al] in which authors use these time criteria, but, as explained in the previous response, these are not the criteria applied by us. For this reason, we have considered it appropriate to maintain studies with 1 year of follow-up.
Hu Z, Sun J, Li R, Wang Z, Ding H, Zhu T, Wang G. A Comprehensive Comparison of LRYGB and LSG in Obese Patients Including the Effects on QoL, Comorbidities, Weight Loss, and Complications: a Systematic Review and Meta-Analysis. Obes Surg. 2020 Mar;30(3):819-827.
Comment 3: The authors mention on page 1 line 44 that they will focus on factors that influence weight loss during bariatric surgery (which ?) in the medium and long term (how long?). However, in the review, they move from mentioning different outcomes of bariatric surgery to outcomes related to weight loss achieved. I think that focusing only on factors that were suggested to explain weight-loss accomplished in gastric bypass vs sleeve gastrectomy after >= 3-5 years would make this review more focused and clear with a more "sharp message".
Response: Thanks for your pertinent commentary.
First of all, we would like to explain that some of the studies cited in the text don’t distinguish among sleeve gastrectomy and Roux-Y-gastric bypass. We have specified some data and highlighted in blue as indicated above. Regarding to time considered medium and long-term, it is explained in response 1.
Comment 4: Though the review is very informative, in some cases it is somehow confusing. I suggest re-organizing each section, so it is easier to follow and come up to conclusions or be able to come up with some "take-home messages". If each section would have been divided into two subsections – one describing information on gastric bypass and one on sleeve gastrectomy + each subsection is further divided according to outcomes (for example in the case of physical activity – from aerobic to resistance training to their combination with regard to weight loss success.
Response: Thanks for your commentary.
We appreciate your recommendation and we have tried to reorganize some sections of the manuscript and to add some conclusions in sections without a final summary just to emphasize the most important messages extracted from the literature reviewed.
Comment 5: Some studies quoted along the text do not include relevant information. For example, when authors mention a result of "bariatric surgery" they should mention which procedure (and preferably for how long was the follow-up). Example on page 7 -row 288: what kind of bariatric surgery? Here also the follow-up is 1 year…
Response: Thanks for this remark.
We have taken it into account and we have clarified these points in the manuscript.
Minor:
Comment 6: Abstract – there is no integrative summary of the main results discussed in the paper. The authors should include some directions and main findings stated in each section.
Response: Thanks for your observation.
We have change the abstract in order to include an integrative summary of the main findings.
Comment 7: Page 1 row 36 is an incomplete sentence.
Response: Thanks for your commentary.
We have rephrased the sentence to make it more understandable in English, it was a translation error. We wanted to say that, despite an initial weight loss after life-style modification programs and/or anti-obesity drugs, it is very common that there is a regain of weight over time.
Comment 8: It is legitimate to focus on studies from the last five years, but it can be helpful to mention why were the past 5 years selected.
Response: Thanks for your commentary.
Bariatric surgery is a field of great interest in our specialty and there is a large number of published studies on the different types of interventions. For this reason, we decided to limit the evidence reviewed to the last 5 years, to assess the most recent studies with surgical techniques that are mainly used today and discard older studies with techniques that have fallen into disuse.
Reviewer 2 Report
The author tried to report collectively this review article and summarized scientific data which could help readers understand integral data about the factors related to the weight loss after bariatric surgery. I would like to recommend that the data should be systematically organized not enumerate a list of a lot of result from research. Most factors have not been fully discovered and investigated and further clinical trials are required to achieve the factors on weight loss after bariatric surgery. It would be better that this review article focuses on and narrow down to some specific factor on weight loss after bariatric surgery.
Author Response
Dear Editor and Reviewers,
We would like to thank you very much for your constructive comments and suggestions which have undoubtedly helped us to improve our manuscript.
In this new version of the manuscript, we have taken into consideration the comments and suggestions from reviewers and have revised the paper accordingly. We have responded to each of the reviewers' comments and have edited the manuscript to address all the reviewers' issues and suggestions.
We have provided the replies to the comments in the following section and have highlighted changes in the manuscript in blue.
We hope that our revised manuscript may now be found acceptable for publication in the journal. Nevertheless, we are of course willing to revise it further according to any other suggestions or concerns raised by the Editor or the Reviewers.
Yours faithfully,
María Molina-Vega, Ana María Gómez-Pérez, and coauthors
Reviewer 2
The author tried to report collectively this review article and summarized scientific data which could help readers understand integral data about the factors related to the weight loss after bariatric surgery.
Comment 1: I would like to recommend that the data should be systematically organized not enumerate a list of a lot of result from research. Most factors have not been fully discovered and investigated and further clinical trials are required to achieve the factors on weight loss after bariatric surgery. It would be better that this review article focuses on and narrow down to some specific factor on weight loss after bariatric surgery.
Response: First of all, we thank the reviewer for his/her comments and for offering us a constructive review of our manuscript.
Together with the comments of Reviewer 1, we have modified the manuscript in order to achieve a better organization and a more understandable manuscript, being more specific in the offered information and clarifying some important points such as surgical technique and follow-up period in all referenced studies. We hope now it to be more adequate.
Round 2
Reviewer 1 Report
Thank you for the revised manuscript.
Reviewer 2 Report
The article has been better edited and rewrittened. If the article focuses on more specific factors and organizes them in terms of weight regain after bariatric surgery, it would be better for readers to understand the reason and make them favor the article